# *Spodoptera frugiperda* Uses Specific Volatiles to Assess Maize Development for Optimal Offspring Survival

**DOI:** 10.3390/insects16060592

**Published:** 2025-06-04

**Authors:** Hanbing Li, Peng Wan, Zhihui Zhu, Dong Xu, Shengbo Cong, Min Xu, Haichen Yin

**Affiliations:** 1Central China Key Laboratory of Integrated Pest Management, Ministry of Agriculture and Rural Affairs, Hubei Key Laboratory of Major Crop Diseases, Pests and Weeds Prevention and Control, Institute of Plant Protection and Soil Fertilizer, Hubei Academy of Agricultural Sciences, Wuhan 430064, China; 18331202139@163.com (H.L.); wanpenghb@126.com (P.W.); ztb799@163.com (D.X.); congshengbo@163.com (S.C.); xumin94@mails.ccnu.edu.cn (M.X.); 2Hubei Insect Resources Utilization and Sustainable Pest Management Key Laboratory, College of Plant Science & Technology, Huazhong Agricultural University, Wuhan 430070, China; zhihui@mail.hzau.edu.cn

**Keywords:** (+)-camphor, growth stage, host recognition mechanism, maize, p-xylene, *Spodoptera frugiperda*

## Abstract

The fall armyworm, *Spodoptera frugiperda*, is a highly destructive pest that severely damages maize crops worldwide. It has developed resistance to chemical pesticides, raising environmental concerns. This study explored how female armyworms use specific maize odors to select optimal sites for laying eggs, with the aim of developing eco-friendly pest control methods. We found that females preferred maize at the seedling stage, guided by two key odors, p-xylene and (+)-camphor, which are more abundant in young plants. These odors strongly attracted females, helping them select sites where their larvae survive better, grow faster, and reproduce more effectively than those on older plants. Our findings suggest that targeting the seedling stage with odor-based traps could effectively control this pest, especially during autumn maize production, when pest populations peak. By identifying these odor cues, this research lays the foundation for sustainable pest management strategies, offering farmers a safer and more environmentally friendly alternative to harmful pesticides while protecting vital maize crops.

## 1. Introduction

*Spodoptera frugiperda*, a member of the family Noctuidae, is a highly destructive polyphagous and migratory pest native to tropical regions of the Western Hemisphere [1,2,3]. Due to its strong flight capability, it has rapidly spread to more than 40 sub-Saharan African countries following its invasion of Nigeria in 2016 [4]. After its initial detection in Yunnan Province, China, in late 2018 [5,6,7], this insect expanded its range across 26 provinces by September 2019 [8,9]. *S. frugiperda* exhibits high reproductive potential, environmental adaptability, explosive population growth, and rapid dispersal. Its host range includes over 300 plant species, with major economically important crops such as maize, rice, peanuts, soybeans, and alfalfa [10,11]. Genetic studies have identified two distinct biotypes, commonly designated as the rice and maize biotypes [12], which, despite their morphological similarity, both preferentially feed on maize [13].

Currently, the management of *S. frugiperda* predominantly relies on chemical control [14]. However, intensive pesticide use has resulted in significant drawbacks, including residue accumulation, environmental pollution, and non-target effects on natural enemies [15]. Furthermore, this pest has developed resistance to multiple insecticides, including parathion, trichlorfon, mevinphos, and permethrin [16]. Compared with chemical pesticides, trapping systems using sex pheromones or food-based attractants offer an efficient and eco-friendly alternative for pest control [17,18]. Recent research has elucidated several sex pheromones of *S. frugiperda* [19]. Unlike sex pheromones, food attractants offer the distinct advantage of directly targeting females, such as geranyl acetate and (Z)-3-hexenyl acetate [20,21], thereby enhancing control efficacy. However, the availability of practical food attractants remains scarce. The development of effective food attractants requires the identification of key odor cues for host recognition. In this study, we collected volatiles emitted by maize plants at four growth stages: seedling stage (SS), small trumpet stage (STS), flowering stage (FS), and milky stage (MS). We evaluated the behavioral preferences of *S. frugiperda* and identified electrophysiologically active components using gas chromatography–electroantennogram detection (GC-EAD), gas chromatography–mass spectrometry (GC-MS), and single sensillum recording (SSR). Then, a Y-tube olfactometer was employed to verify the olfactory responses to these active compounds, enabling the identification of critical host-recognition cues.

In addition, understanding the pest preference for host plants at different growth stages is crucial for developing effective management strategies [22]. However, few studies have examined the growth performance of *S. frugiperda* larvae on maize at different phenological stages, and the mechanism by which females assess host growth stages remains unclear. Therefore, we compared relative concentrations of key odor cues in different growth stages and evaluated the growth performance of *S. frugiperda* larvae on maize at these stages to assess how adult oviposition choices affect offspring survival. This study may provide a theoretical foundation for determining the optimal control period and developing food attractants.

## 2. Materials and Methods

### 2.1. Test Materials

*S. frugiperda* larvae were collected from corn fields around the Ezhou Experimental Base of the Hubei Academy of Agricultural Sciences (114° 39′ E, 30° 23′ N) in June 2023. The larvae were individually reared in 24-well plates containing an artificial diet until pupation under laboratory conditions. Pupae were subsequently transferred to glass culture dishes and placed in an oviposition cage lined with wax paper. Adult moths were maintained at 26 ± 1 °C with a 14:10 h (L:D) photoperiod and 60 ± 5% relative humidity in climate-controlled chambers. Adults were provided with 10% honey solution via cotton wicks, which were replaced daily. Egg masses were collected following oviposition, and newly hatched first instar larvae from synchronized cohorts were used for subsequent experiments.

The artificial diet was prepared according to Li et al. [23], consisting of soybean powder, yeast powder, wheat germ, ascorbic acid, benzoic acid, sodium benzoate, edible oil, agar, acetic acid, and purified water.

Maize plants (*Zea mays* cv. Zhengdan 958, supplied by Hefei Fengzhong Seed Industry) were cultivated to four growth stages for experiments. The standards for identifying the growth stages were as follows:

SS stage: 3–4 leaf stage, the length of plants is 2 to 20 cm.

STS stage: 7–10 leaf stage, leaf number index is about 46.

FS stage: pollen and filaments appeared.

MS stage: the kernels are yellow and release milky juice when punctured.

### 2.2. Experimental Instruments

Electrophysiologic responses of insects were measured using a gas chromatography–electroantennogram detection (GC-EAD) system consisting of an Agilent 7820 GC (Agilent Technologies, Santa Clara, CA, USA) and a Syntech IDAC-2 electroantennogram detector (Syntech Research, Gewerbestr. 3, 79256 Buchenbach, Germany), supplied by Tangshan Dinggan Technology Company (Tangshan, China). Plant volatile components were determined using a GC-MS system, specifically RACE GC 2000 (Shimadzu Corporation, Kyoto, Japan). Behavioral responses were assayed using a Y-tube olfactometer with a main trunk (Φ = 18 cm × 4 cm) and two arms (Φ = 14 cm × 4 cm) angled at 75°. The arms were connected via Teflon tubing to an odor source bottle, a flowmeter, a filtration device, and an oil-free vacuum pump.

### 2.3. Collection of Volatiles from Maize

Volatile compounds were collected from maize plants at different growth stages using a dynamic headspace sampling system. The system consisted of a transparent glass collection chamber (Φ = 30 cm × 50 cm) containing potted plants and two glass ports: one inlet equipped with an activated charcoal filter to purify incoming air and one outlet connected to a volatile collection trap.

The collection trap (Φ = 0.7 cm × 15 cm) was packed with 200 mg of Super Q adsorbent (80/100 mesh, Waters Corporation, Milford, MA, USA). Continuous sampling was performed for 15 h at ambient temperature with a purified airflow maintained at 300 mL/min (optimized based on preliminary tests).

Following collection, trapped volatiles were eluted using 500 μL HPLC-grade n-hexane (Sigma-Aldrich, St. Louis, MO, USA). The eluate was immediately transferred to amber glass vials and stored at 4 °C until analysis (typically within 24 h). All glassware was heat-sterilized at 180 °C for 4 h prior to use to minimize contamination.

### 2.4. Behavioral Bioassay of S. frugiperda

Behavioral responses of both male and female adults of *S. frugiperda* were assayed using the Y-tube olfactometer. All tests were conducted in a darkroom at 4:00 p.m. to coincide with the activity period of the insects. The Y-tube was positioned in a ventilated chamber, and two 1 cm^2^ filter paper strips, each coated with volatile samples, were placed in the odor source bottle. Before the experiment, insects were dark-adapted and starved for 4 h. For the bidirectional choice assay evaluating volatiles from different growth stages, 10 insects were introduced into the Y-tube each time. Each treatment group was tested across 10 replicates (total n = 100 insects). After 45 min, the number of insects in each arm was recorded. To minimize external interference, chamber ventilation was maintained, and all ambient light sources were turned off throughout the experiment. Between tests, the inner walls of the Y-tube were cleaned with 75% ethanol and dried.

### 2.5. Oviposition Selectivity of S. frugiperda Females

The oviposition selection behavior of *S. frugiperda* females on maize leaves at different growth stages was assessed using a two-choice assay (Figure 1). The experimental setup consisted of two insect cages, each containing fresh maize leaves from different growth stages, connected by a transparent cylindrical tube. Each cage had its own supply of degreased cotton soaked in 10% honey solution, which was replenished daily.

Mated female *S. frugiperda* were obtained by pairing newly emerged individuals in disposable cups provided with degreased cotton soaked in 10% honey solution. Once females began ovipositing in these cups, they were considered mated and used for the assay.

Tested adults and leaves were replenished daily at 08:00 am. Observations were recorded after 24 h (at 08:00 a.m. the following day) before starting the subsequent experimental trial. For each replicate, a mated female was released into the center of the transparent cylindrical tube connecting two cages. The female could then choose to enter one of the two cages, each containing maize leaves from a different growth stage. The leaves were trimmed, and their bases submerged in water to prevent desiccation. Each female was observed for a set period, and then eggs were recorded. Egg masses deposited on the leaves (excluding those on the cage walls) were collected and quantified daily. The experiment was conducted with 20 replicates (n = 20 females).

### 2.6. Growth Performance of S. frugiperda Colonized on Maize at Different Growth Stages

Newly hatched *S. frugiperda* larvae were fed on maize leaves from the SS, STS, FS, and MS stages. Each growth stage was considered a treatment group consisting of three replicates, each with 20 newly hatched larvae (total n = 60 per growth stage). Fresh leaves were replaced daily until pupation. The following parameters were recorded daily: developmental duration of each life stage, pupal duration, survival rate, larval body weight (measured on days 10 and 15), pupal weight, and pupation rate. Newly emerged healthy males and females were paired in a 1:1 ratio within disposable cups, and daily egg production and egg hatching rates were recorded.

### 2.7. GC-EAD and GC-MS Analysis of Volatile Compounds from Maize at Key Growth Stages

Electrophysiologically active volatiles from maize were identified by GC-EAD and GC-MS. Two glass capillaries (4–5 cm in length) filled with 0.9% sodium chloride solution were used as measuring electrodes and reference electrodes. Two clean silver wires (4.5 cm in length) served as electrode leads, with each wire inserted into one capillary. The assembled electrodes were secured to the measuring stage and reference pole of the PRG-3 electrode holder.

Antennae from newly emerged females of *S. frugiperda* were excised at the base, and the distal tip was slightly trimmed. The reference electrode of the PRG-3 setup was removed, and the antennal base was stabilized through the surface tension from the saline solution. The antennal tip was connected to the measuring electrode under a stereomicroscope.

For volatile analysis, 3 μL of plant volatiles were injected into the gas chromatograph inlet. GC-EAD responses were recorded, and active peaks were identified by matching their retention times with electrophysiological responses. Compounds were tentatively identified using the NIST11 mass spectral library and confirmed by comparing retention times and mass spectra with those of authentic standards. The chromatographic column was DB-5MS (30 m × 0.25 mm × 0.50 μm, Agilent Technologies, Santa Clara, CA, USA). Data acquisition and analysis were performed using GC-EAD 2014 v1.2.5 software.

### 2.8. SSR Validation of GC-EAD Results

The electrophysiologically active compounds identified through GC-EAD analysis were subsequently validated by the SSR technique. Newly emerged female *S. frugiperda* moths (24–48 h post-eclosion) were immobilized on glass slides using dental wax grooves under a stereomicroscope. The head and antennae were carefully positioned outside the groove while maintaining proper physiological conditions. The insect body was securely fixed to prevent movement. A reference electrode was inserted into the compound eye, and a recording electrode was positioned at the base of the antennal sensilla. An odor delivery system had its outlet tube positioned 1 cm from the antenna. Test odorants (10 μL aliquots) were applied to filter paper strips (1 cm^2^) and presented at intervals of ≥30 s to allow complete neuronal recovery between stimuli. Under the microscope, about 20 olfactory sensors distributed from the tip to the base in one antenna were randomly selected to be stimulated with the agent, and 10 antennae were employed in this test. Sensors without electrophysiological signals were ignored.

### 2.9. Olfactory Responses of Female S. frugiperda to Electrophysiologically Active Components

Behavioral responses of female *S. frugiperda* to the electrophysiologically active components were identified by the Y-tube olfactometer, with liquid paraffin used as control (CK). The test method was the same as described in Section 2.4.

### 2.10. Data Analysis

Behavioral preference assays of *S. frugiperda* were subjected to rigorous statistical evaluation employing Pearson’s chi-square test (α = 0.05) implemented in SPSS Statistics 26. Developmental and reproductive performance metrics across different maize growth stages were analyzed through the *LSD* test (*p* < 0.05). Electrophysiological waveforms recorded during SSR were processed using AutoSpike V3.9.0. Neuronal responsiveness was quantified by comparing action potential frequencies in one-second epochs before and after stimulus administration. All graphical representations were generated using GraphPad Prism 8.0.

## 3. Results

### 3.1. Behavioral Bioassay of S. frugiperda

Our preliminary tests showed that volatiles from maize at all growth stages elicited significant attractive responses in adults *S. frugiperda* when compared with the blank control. In order to compare the preference of *S. frugiperda* from different growth stages of maize, no blank control was set up in this experiment.

Female *S. frugiperda* exhibited a distinct preference hierarchy for maize volatiles across four growth stages: SS > STS > FS > MS. In pairwise comparisons using a Y-tube olfactometer, SS volatiles attracted significantly more females than STS volatiles (35 vs. 20; χ^2^ = 8.18, *p* < 0.01). Similarly, FS volatiles were preferred over MS volatiles (20 vs. 6; χ^2^ = 15.08, *p* < 0.01), and STS volatiles were more attractive than FS volatiles (27 vs. 8; χ^2^ = 20.63, *p* < 0.01) (Figure 2).

In contrast, male *S. frugiperda* showed no significant preference for maize volatiles at different growth stages (*p* > 0.05) (Figure 3).

### 3.2. Oviposition Selectivity of Female S. frugiperda

Female *S. frugiperda* showed a clear hierarchy in oviposition preference for maize at different growth stages: SS > STS > FS > MS. In two-choice assays, the average number of eggs laid on SS leaves was significantly higher than on STS leaves (405.2 vs. 196.7; χ^2^ = 144.47, *p* < 0.01). Similarly, STS leaves were preferred over FS leaves (332.5 vs. 117.5 eggs; χ^2^ = 205.44, *p* < 0.01). FS leaves were preferred over MS leaves (153.7 vs. 0 eggs; χ^2^ = 307.33, *p* < 0.01), thus demonstrating a strong preference for earlier growth stages, particularly SS. Moreover, the trend in selective preference of adults for plants at different growth stages was consistent with the results in Section 3.1, while the response rates were all above 95%, higher than those reported in Section 3.1. This demonstrates that host plants displayed a stronger attractive effect than their volatiles alone (Figure 4).

### 3.3. Effects of Feeding Maize Leaves at Different Growth Stages on the Growth, Development, and Reproduction of S. frugiperda

The survival rates of *S. frugiperda* larvae varied significantly across maize growth stages (Figure 5A). The highest survival rate was observed on SS leaves (91.07%), followed by FS (77.97%) and STS (73.33%). Complete mortality occurred within 5 days on MS leaves, leading to their exclusion from subsequent analyses.

Larvae fed on SS leaves had significantly shorter developmental durations of larval stages (16.33 days) compared to other stages (*p* < 0.05; Figure 5B). And the larvae fed on SS leaves had the shortest developmental durations of pupal stages (9.09 days).

On day 10, there was no significant difference in body weight between larvae fed on SS (0.13 g) and FS (0.11 g) leaves (*p* > 0.05). However, by day 15, larvae fed on SS leaves exhibited significantly higher body weights and pupal weights compared to those fed on other stages (*p* < 0.05; Figure 5D,E) with the highest pupation rate (91%; Figure 5F).

The reproductive performance showed parallel trends, with females from SS- and STS-reared producing over 1700 eggs per female and hatching rates exceeding 90%, significantly surpassing FS-reared females (965 eggs with 72% hatching; *p* < 0.05; Figure 5G,H).

### 3.4. GC-EAD and GC-MS Analysis of Volatile Compounds from Maize at Key Growth Stages

Given the strong oviposition preference and superior larval performance on SS leaves (Section 3.2 and Section 3.3), we focused on volatile compounds from SS leaves for GC-EAD and GC-MS.

GC-EAD analysis revealed significant antennal responses at retention times of 8.43 and 16.08 min (Figure 6). Subsequent GC-MS identification confirmed that peaks at 8.43 and 16.08 min corresponded to p-xylene and (+)-camphor, respectively, by comparing with authentic standards. Therefore, p-xylene and (+)-camphor were selected for further behavioral studies (Figure 7).

Only one data of FID peaks is shown as representative in this test due to its stable repeatability, the other FID peaks are shown in Appendix A.

A quantitative comparison of volatile profiles across different growth stages revealed that the concentration of p-xylene in SS leaves (18.2% relative content) was significantly higher than in other periods. The concentration of (+)-camphor in SS was also the highest (2.48%) although it showed no significant differences compared with those in STS (1.30%), FS (2.08%) and MS (1.78%) (Table 1).

### 3.5. SSR Validation in GC-EAD Results

SSR was performed to validate the electrophysiological responses to p-xylene and (+)-camphor identified via GC-EAD. Female *S. frugiperda* adults were tested with p-xylene at 5%, 10%, and 20% dilutions in liquid paraffin and (+)-camphor at 1%, 5%, and 10%. According to the differences in the amplitude of the action potential (spike), two types of olfactory receptor neurons (ORN a and ORN b) were recorded in the middle receptors of the antennae (Figure 8).

For p-xylene:

Neuron a showed excitation at 5% (13.00 spikes/s) and 20% (12.00 spikes/s) but inhibition at 10% (−8.00 spikes/s) compared to the paraffin control (0.67 spikes/s).

Neuron b showed dose-dependent excitation with the highest response at 20% (80.33 spikes/s vs. control: 6.00 spikes/s).

For (+)-camphor:

Neuron a had the strongest response at 5% (15.00 spikes/s), with lesser responses at 1% (9.33 spikes/s) and 10% (3.33 spikes/s).

Neuron b showed a biphasic response: inhibition at 5% (−3.67 spikes/s) and neuronal responses at 1% and 10% (4.33 spikes/s). These responses confirm that female *S. frugiperda* have specific olfactory neurons to p-xylene and (+)-camphor (Figure 9).

### 3.6. Olfactory Response of Female Adults of S. frugiperda to Volatile Active Components

Both GC-EAD and SSR confirmed electrophysiological responses of female *S. frugiperda* adults. We evaluated their behavioral effects on female *S. frugiperda* using a Y-tube olfactometer. For p-xylene, females showed significant dose-dependent attraction at all concentrations, with the strongest effect at 20% (χ^2^ = 37.38, df = 1, *p* < 0.01) (Figure 10A).

For (+)-camphor, all concentrations (1%, 5%, and 10%) attracted females significantly more than paraffin control (Figure 10B).

## 4. Discussion

In this study, we used GC-EAD and SSR to demonstrate that *S. frugiperda* exhibits significant electrophysiological responses to p-xylene and (+)-camphor. Notably, the concentration of these two compounds varies across different growth stages of maize, allowing female adults to select optimal oviposition sites that enhance offspring fitness.

The co-evolution of herbivorous insects and their host plants has led to an intricate relationship where host plant volatiles play a crucial role in influencing insect behaviors, such as oviposition and feeding, in the long-term evolution process [24,25]. Our findings align with previous research, which has shown that p-xylene and (+)-camphor derivatives are attractive to various insect species. For example, p-xylene has been reported to attract *Adelphocoris lineolatus* [26], and (+)-camphor has been found to elicit attraction in *Pagiophloeus tsushimanus* [27]. Additionally, studies have demonstrated that p-xylene from *Luffa acutangula* L. induces differential responses in male and female *Zeugodacus cucurbitae* (Coquillett) [28], while m-xylene strongly attracts *Apolygus lucoru* in field experiments [29]. Furthermore, (−)-camphor has been shown to attract *Monochamus saltuarius* in indoor behavioral assays [30]. However, there are limited reports about the attractive effects of p-xylene and (+)-camphor on *S. frugiperda*. In this study, our data confirmed these as important odor cues for host identification and oviposition site selection by this insect.

The selection of oviposition sites by female directly determines the habitat and subsequent development success of their offspring. Our study revealed that female *S. frugiperda* preferred to lay eggs on maize leaves at the seedling stage, where larval survival rates, developmental speed, and weight gain were significantly higher compared to later growth stages. Previous investigations found that maize sowing times vary across production systems, with differences in soil conditions contributing to asynchronous seedling emergence [31,32]. Consequently, adjacent maize fields often exhibit concurrent but distinct growth stages during cultivation, rendering the host selection behavior of *S. frugiperda* between these stages a critical research focus. When female insects oviposit on tender growth stages, such as the SS stage, the resulting larvae may face limited food resources due to the small leaf surface area. To compensate, larvae may disperse to adjacent plants through silk-spitting behavior [33]. However, due to their limited mobility, larval survival ultimately depends on the oviposition sites selected by females. The oviposition preferences of *S. frugiperda* reflect a trade-off between the quantity and quality of food. Our study indicates that despite the potential for limited food resources, females still prefer to choose high-quality food sources, such as young leaves with rich nutrients, and exhibit lower levels of defensive secondary metabolites [34,35,36]. Previous studies have reported that as plants mature, they accumulate secondary metabolites that can deter or impair insect herbivores [35]. For example, Ryu et al. [37] observed gradual increases in flavonols and aflavins in the leaves of *Cordyceps sinensis* with plant age, while Ghasemzadeh et al. [38] found the contents of total flavonoids and total phenols peak in 9-month-old maize rhizome extracts. Therefore, we speculated that mature maize, with higher levels of insect-resistant substances and lower nutrient content, is less suitable for larval growth and development, explaining the observed differences across growth stages. Given that p-xylene and (+)-camphor are more abundant in seedling-stage maize, female *S. frugiperda* can use these cues to select this optimal stage for oviposition, thereby enhancing offspring fitness. This finding has significant implications for pest management, as it suggests that control strategies should prioritize the seedling stage, particularly during autumn maize production. Focusing on this critical stage could maximize the efficacy of interventions, given the large population base at this stage.

Our study also highlighted gender-specific responses to host volatiles, with female *S. frugiperda* exhibiting distinct preferences compared to males, a pattern consistent with observations in other insect species. For example, in *Heliothis virescens*, females show a weaker response to the main sex pheromone compound Z11-16:Ald compared to males [39], and in *Grapholita molesta*, males are significantly attracted to female-released sex pheromones (Z8-12:Ac and E8-12:Ac). These gender-specific olfactory responses suggest that food attractants, which target host recognition cues, might be particularly effective for managing female *S. frugiperda*, as they are the primary agents in selecting oviposition sites. Currently, while sex attractants have been developed for *S. frugiperda* control [40], there is a relative scarcity of food attractants. Our study provides a theoretical foundation for developing such attractants by identifying key host recognition cues, potentially offering a more sustainable and targeted approach to pest management.

In summary, this study investigated the host preference and recognition mechanism of *S. frugiperda*. The odor cues identified in this study can be developed for field trapping. While our findings are promising, field validation is necessary to confirm their efficacy in practical pest control scenarios. Future research should focus on determining the optimal combination ratios of these two compounds to maximize their attractiveness, as previous studies have shown that the combination of attractive compounds in a certain proportion can significantly enhance their effects [41,42]. Additionally, exploring the integration of these food attractants with sex attractants or visual cues could lead to more effective strategies for monitoring and controlling *S. frugiperda* populations [43,44]. In this study, it was found that *S. frugiperda* showed lower response rates when faced with the selection between volatiles in behavioral bioassays, while the response rates increased substantially in the oviposition preference test. The possible reason might be that in the experiment of oviposition preference, the host plants were real and not merely the volatile substances simulating their odors. Obviously, the real host plants provided abundant cues, thus demonstrating a stronger attraction than the sole olfactory cues [45]. This result indicated that there might be other factors besides olfactory cues influencing the host recognition of *S. frugiperda*, which can be further studied in the future. Therefore, the comprehensive utilization of multiple clues can improve the trapping efficiency. Moreover, our attractants can also be used in combination with repellents as part of a push–pull strategy [46]. Such integrated approaches could provide a feasible method for field monitoring and control, addressing the current gap in food attractant development and enhancing the sustainability of pest management practices.

## 5. Conclusions

This study demonstrated that the maize volatiles p-xylene and (+)-camphor mediate *S. frugiperda* host-stage recognition and oviposition site selection. Meanwhile, the choice of oviposition sites by adults also significantly affected the growth of the offspring larvae.

## Figures and Tables

**Figure 1 insects-16-00592-f001:**
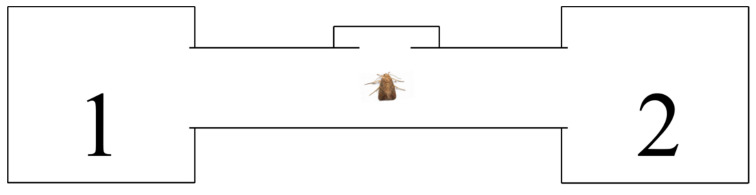
Experimental device for oviposition selection of *S. frugiperda.* Note: The number in the figure is the position of the leaves.

**Figure 2 insects-16-00592-f002:**
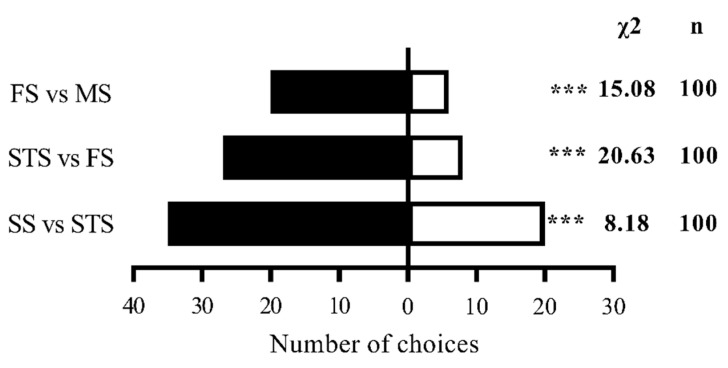
Olfactory selection of female *S. frugiperda* to maize volatiles at different stages. Note: In the histogram, *** indicates a significant difference (*p* < 0.01) (chi-square test). n represents the total number of tested adults.

**Figure 3 insects-16-00592-f003:**
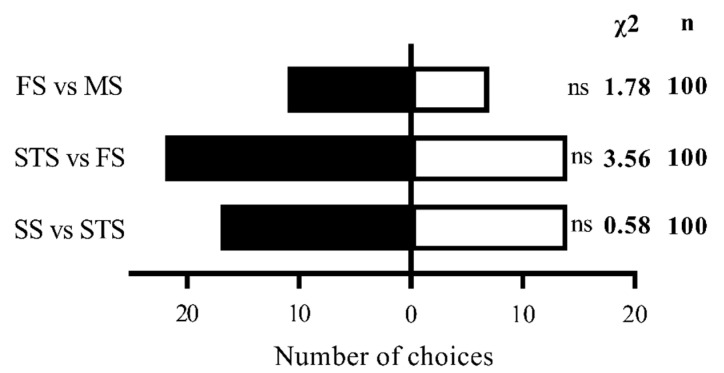
Olfactory selection of male *S. frugiperda* to maize volatiles at different stages. Note: ns in the histogram indicates that the difference is not significant (*p* > 0.05) (chi-square test). n represents the total number of tested adults.

**Figure 4 insects-16-00592-f004:**
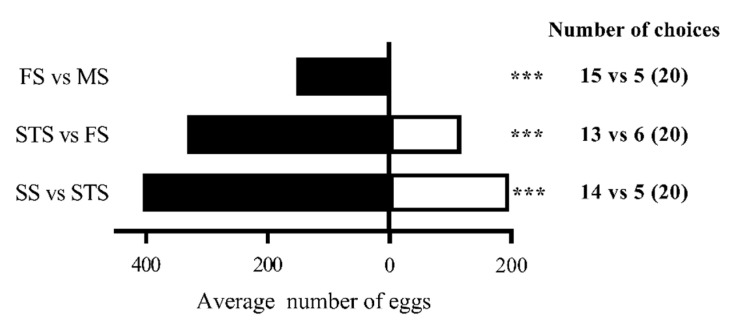
Oviposition selectivity of female *S. frugiperda* to maize at different growth stages. Note: Number of choices represents the number of adults selected FS vs. MS, STS vs. FS, and SS vs. STS. The numbers in parentheses represent the total number of test insects for each treatment. *** in the histogram indicates that the difference is significant (*p* < 0.01) (chi-square test).

**Figure 5 insects-16-00592-f005:**
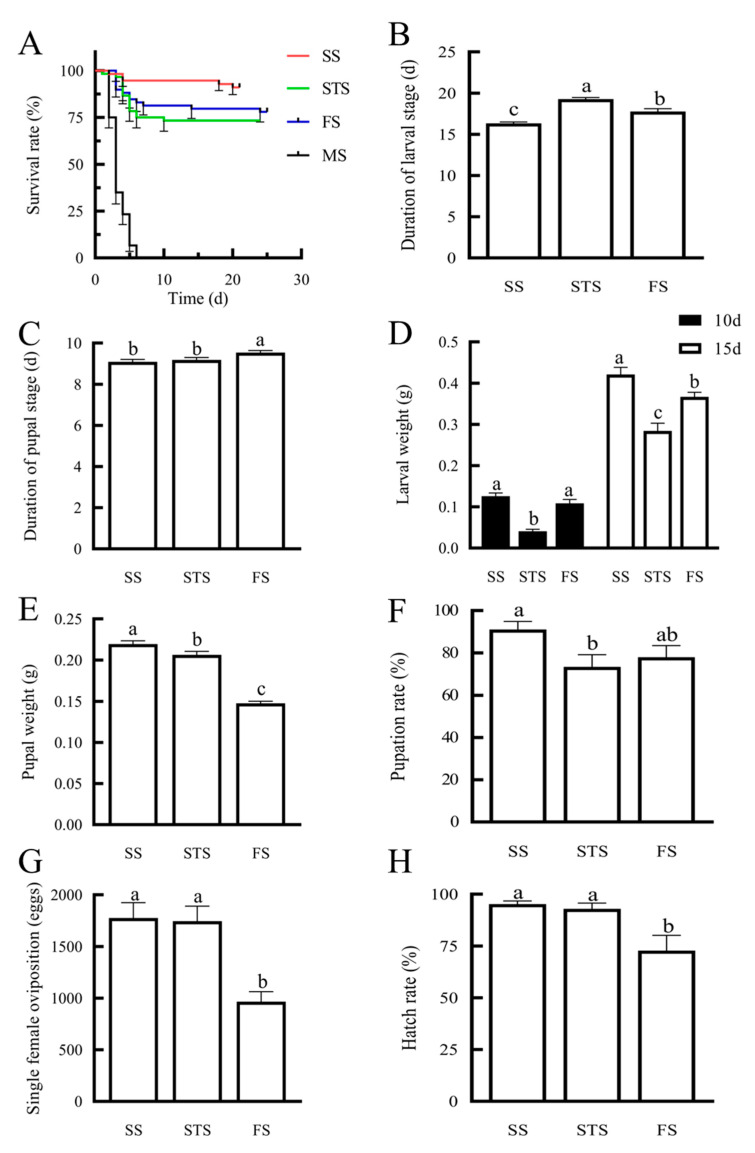
Effects of feeding on maize leaves at different growth stages on the growth, development, and reproduction of *S. frugiperda*: (**A**) survival rate; (**B**) duration of the larval stage; (**C**) duration of the pupal stage; (**D**) larval weight; (**E**) pupal weight; (**F**) pupation rate; (**G**) single female oviposition; (**H**) hatch rate. Note: Different letters in the graph indicate that there are significant differences between different treatments (*p* < 0.05) (LSD test) (mean ± standard error).

**Figure 6 insects-16-00592-f006:**
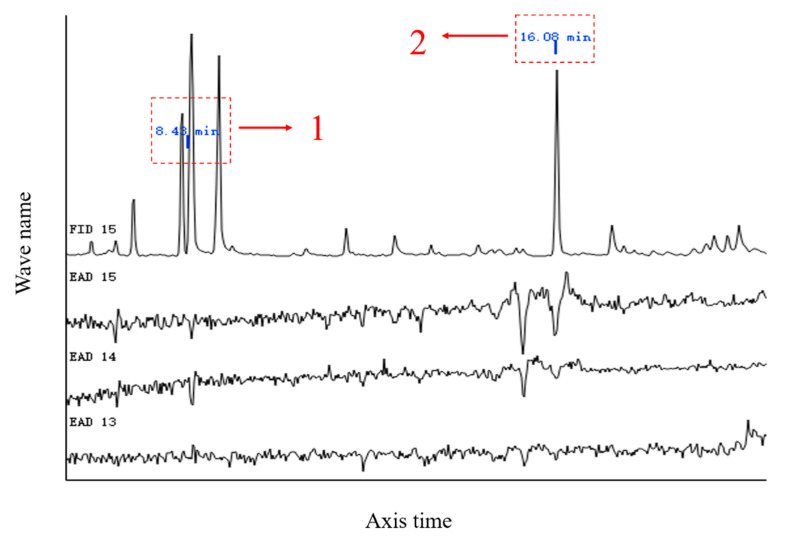
GC-EAD response of female *S. frugiperda* to maize volatiles. Note: The numbers in the figure are two active components of the SS maize volatiles.

**Figure 7 insects-16-00592-f007:**
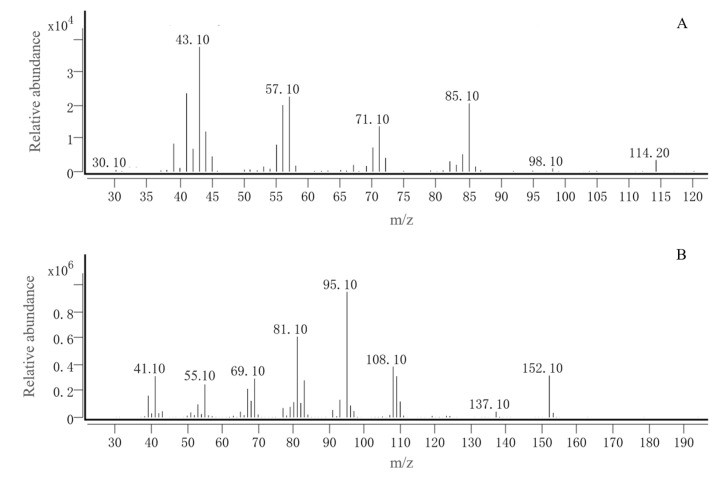
Mass spectra of No. 1 and No. 2 compounds in the corn volatiles (**A**): p-xylene; (**B**): (+)-camphor.

**Figure 8 insects-16-00592-f008:**
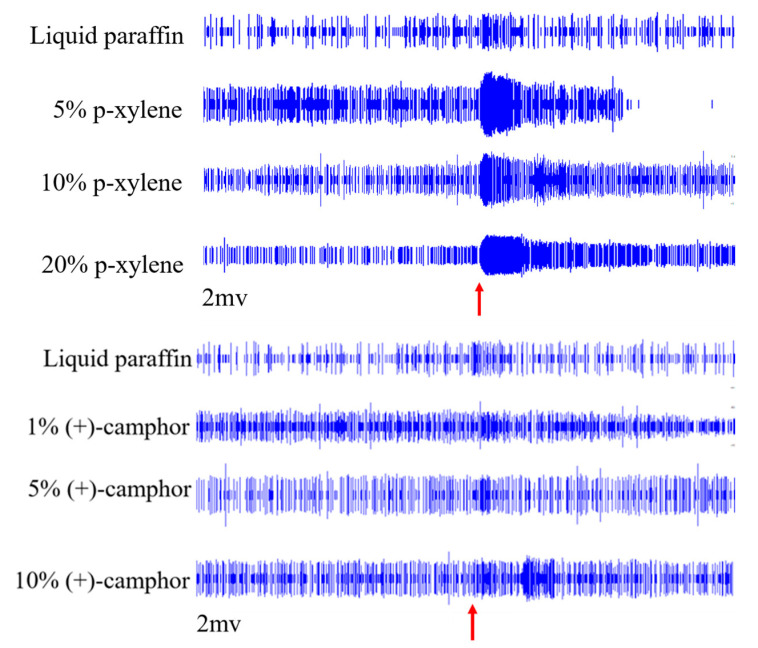
Neuronal response of one antennal sensilla of *S. frugiperda.* Note: The red arrows indicate the position of the stimulus.

**Figure 9 insects-16-00592-f009:**
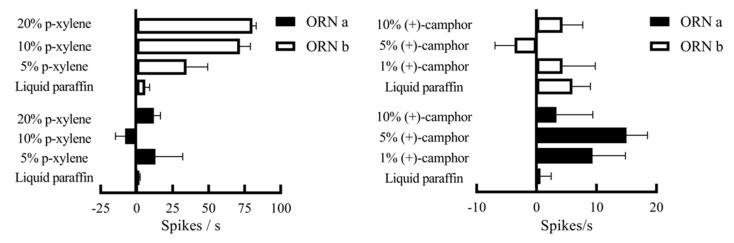
Responses of different neurons in the antennal sensilla of *S. frugiperda* to volatile active components of the host. Note: ORN a and ORN b in the figures represent two kinds of neurons. The spike difference of 1 s before and after stimulation was counted, and the error line was the standard error.

**Figure 10 insects-16-00592-f010:**
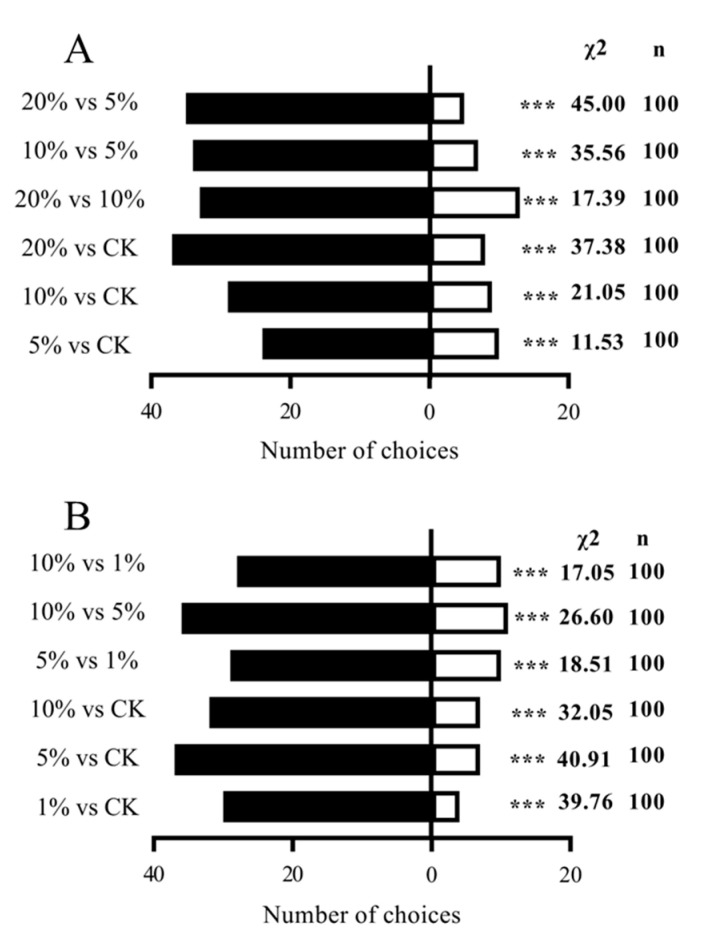
Olfactory behavior of *S. frugiperda* female adults in response to three different concentrations of compounds: (**A**) p-xylene; (**B**) (+)-camphor. Note: *** indicates a significant difference (chi-square test) (*p* < 0.01). n represents the total number of tested adults.

**Table 1 insects-16-00592-t001:** The relative content percentage of compounds in each growth period.

No.	Volatile Components	CAS No.	Relative Content (%)			
			SS	STS	FS	MS
1	p-xylene	106-42-3	18.20 ± 1.04 a	0.96 ± 0.28 b	0.15 ± 0.08 b	0.68 ± 0.27 b
2	(+)-camphor	464-49-3	2.48 ± 0.93 a	1.30 ± 0.06 a	2.08 ± 0.44 a	1.78 ± 0.25 a

Note: Different letters in the table indicate that there are significant differences between different treatments (*p* < 0.05) (LSD test) (mean ± standard error).

## Data Availability

The original contributions presented in this study are included in the article/Appendix A. Further inquiries can be directed to the corresponding author.

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
