# Peer review of "Spodoptera frugiperda Uses Specific Volatiles to Assess Maize Development for Optimal Offspring Survival"

_insects, 2025, doi:10.3390/insects16060592_

Round 1
Reviewer 1 Report
Comments and Suggestions for Authors
General comments:
The paper reports an exhaustive set of laboratory assays on electrophysiological and behavioral responses of Spodoptera frugiperda to plant volatiles.
The introduction presents the integrated pest management context of the research although it only mentions the use of plant volatiles in general terms, leaving for the discussion the mention of specific references on specific compounds or examples. Considering the focus of the paper, it would benefit the introduction to include some more information on the effectiveness of plant volatiles in IPM and a short relation of the most relevant components that have been discovered so far.
The experimental design is thorough, using different techniques to identify key compounds and test behavioral responses in two-way choice assays and electrophysiological discovery and validation by means of GC-EAD and SSR.
Results are clearly presented and discussed, and methodology is mostly explained in sufficient detail.
In the discussion, the authors cite several references to other pest species attracted to the plant volatiles identified in this study, and they explore the implications of these findings for integrated pest management (IPM) strategies. Given the applied nature of the research, it would be relevant to highlight these implications in the conclusions.
Overall, I believe that the work is of interest in the field, and I congratulate the authors on it.
There is only some specific issues that I’d like to raise to improve the quality of the publication.
Specific comments:
Line 64: final point for the sentence is missing. Is the sentence complete?
Lines 156-173: The details of the chromatographic column are not reported. Please do so.
Lines 174-184: SSR validation experiments should include some detail on the sequence of stimulus presentation on every antenna, stating if randomization of the order was applied.
Figure 4. The y axis legend of B and C is not completely clear and the legend does not help to clarify. I suggest ‘larval stage duration’ and ‘pupal stage duration’ either in the axis or in the legend. Note that ‘larval stage’ should be used instead of ‘larvae stage’. In D, ‘larval weight’ would also be clearer than just ‘weight’. In H, although ‘egg hatchability’ is not incorrect, I would suggest to use a different term, like ‘Hatch success’ or ‘Hatch rate’.
Figure 5. The legend states that GC-EAD response is for ‘adults’, but the superposition of several traces is understandably manually overlapped. This should be explained, along with the information if the reported traces are exemplary and how were they selected. The FID 15 and EAD15, 14 and 13 is surely the label for different insects, but they seem to be mispositioned. Please correct this. The term ‘ingredient’ seems to indicate a mixed recipe, please use the term ‘active components of the maize plant volatile’ also stating the growth stage of the plant used (SS?)
Author Response
Reviewer 1
Comments 1: Line 64: final point for the sentence is missing. Is the sentence complete?
Response 1: Thank you for pointing this out. We have completed the sentence accordingly in line 69-70.
Comments 2: Lines 156-173: The details of the chromatographic column are not reported. Please do so.
Response 2: Thank you for your kind comments. The detail of the chromatographic column has been added in lines 188-189.
Comments 3: Lines 174-184: SSR validation experiments should include some detail on the sequence of stimulus presentation on every antenna, stating if randomization of the order was applied.
Response 3: Thank you for pointing this out. We have added the details of the SSR verification experiment in lines 201-204.
In the SSR verification, about 20 olfactory sensors distributed from the tip to the base in one antenna were randomly selected to be stimulated with the agent and 10 antennae were employed. Sensors without electrophysiological signal were ignored.
Comments 4: Figure 4. The y axis legend of B and C is not completely clear and the legend does not help to clarify. I suggest ‘larval stage duration’ and ‘pupal stage duration’ either in the axis or in the legend. Note that ‘larval stage’ should be used instead of ‘larvae stage’. In D, ‘larval weight’ would also be clearer than just ‘weight’. In H, although ‘egg hatchability’ is not incorrect, I would suggest to use a different term, like ‘Hatch success’ or ‘Hatch rate’.
Response 4: Thank you for pointing this out. We have revised Figure 4 accordingly.
Comments 5: Figure 5. The legend states that GC-EAD response is for ‘adults’, but the superposition of several traces is understandably manually overlapped. This should be explained, along with the information if the reported traces are exemplary and how were they selected. The FID 15 and EAD15, 14 and 13 is surely the label for different insects, but they seem to be mispositioned. Please correct this. The term ‘ingredient’ seems to indicate a mixed recipe, please use the term ‘active components of the maize plant volatile’ also stating the growth stage of the plant used (SS?)
Response 5: Thank you for pointing this out. There were some repetitions in the GC-EAD experiment, while the FID peaks of these repetition were basically the same. Therefore, we selected one of them as a representative, the other FID peaks are shown in Supplementary File Figure 1. The results of electrophysiological responses with stable baselines and obvious responses were selected for further analysis. We have also modified Figure 5 accordingly.
Comments 6: Considering the focus of the paper, it would benefit the introduction to include some more information on the effectiveness of plant volatiles in IPM and a short relation of the most relevant components that have been discovered so far.
Response 6: Thank you for pointing this out. We have supplemented the previous research findings that geraniol acetate and (Z)-3-hexenyl-acetate in maize volatiles affect the oviposition preference of S. frugiperda in lines 55-61.
Comments 7: In the discussion, the authors cite several references to other pest species attracted to the plant volatiles identified in this study, and they explore the implications of these findings for integrated pest management (IPM) strategies. Given the applied nature of the research, it would be relevant to highlight these implications in the conclusions.
Response 7: Thank you for pointing this out. We propose that these plant volatiles can be developed into attractants for field trapping. Additionally, they may be used in combination with repellents as part of an push-pull strategy. We provided supplementary explanations in lines 423 and 432.
Reviewer 2 Report
Comments and Suggestions for Authors
The paper is actual and up-to-date. The findings of the researchers open up a new way of understanding and combatting lepidopteran pests such as S. frugiperda. This paper suits the journals scope and should be interesting to readers, especially those working in integrated pest management.
This manuscript shows that S. frugiperda use certain plant volatiles to determine what plants are best suited to oviposit on. The researchers showed that S. frugiperda is attracted to the youngest seedlings using several methods, showing that both males and females are attracted to those plants in general as well as that females choose those to oviposit on these plants and have a measurable reaction when exposed to those volatiles.
While the research approach is novel and interesting, the methods and presentation of the results should be thoroughly reworked and made more understandable and consistent. Flaws are listed below, upon revision of those, the paper can be published.
L58-59: The growth stages could be better explained/defined
L100: 75 Degrees.
L114: While the meaning of the sentence is clear, could another word be used instead of baked?
L120: Could ‘coated’ be used here instead of ‘impregnated’?
L121-122: How many insects were used in total and how many in each pairwise comparison?
L124: The method could be explained more precisely, how can each volatile have been tested 10 times, when 2 stages have been tested only once, and two others have been tested two times?
L134: Newly emerged individuals
L135-136: A timeframe could be provided here
L163: Is this supposed to mean females?
L199: Why was there no blank control in this experiment? Figure 9 shows controls for a different experiment.
L201: What was the reason that only adjacent growth stages were compared and not all stages against all others, e.g. MS vs. SS.
L202-204: The number of insects that decided on one or the other side does not add up to groups of 10 insects nor does it add up to 10 insects tested 10 times. Also, the number of insects varies for each comparison. If a considerable number of insects did not decide or had to be exempt for some reason, this needs to be shown in the results.
L206: The X-axis title is misleading/translated wrong – this also applies to the following similar graphics. For all graphics, one side of the bars could be a black outline and white inside to better contrast the two, especially for the graphs in figure 4.
L207: n should be indicated in the caption (in all graphics). Captions for figures 1 and 2 are not concise.
L211: The bars do not need to be elevated compared to figure 1.
L244: Figure A Y-Axis title is inconsistent with other graphs using percentages (F and H). One of the bars in the graphs could be white with black outline to better contrast the graphs, compared to two different shades of grey. Graph D Y-Axis title could be more precise in the graph and caption – worm weight- when other graphs say larval. Graph F uses a scale of 0.0-1.0 for percentage values when other graphs use 0-100, was the emergence rate 1 %?
L255: Can an x-axis be added to this graphic?
L266-269: Could other molecules have increased in percentage, adding to the recorded effect?
L280: Is there an X-Axis to this graphic? Is there an explanation to what the red bars in the graphic are supposed to show the reader?
L293: Figure 8 X-Axis titles are inconsistent (spacing).
L294: a and b do not show up in the graphics. It is confusing to talk about graph a and graph b when both graphs contain ORN a and b. The term ORN is not explained in the text.
L298: the sentence is unnecessary, the previous chapters showed this.
L306: Figure 9 – the term CK is not explained in the text
L311: repetitive
L314-315: is there any indication that there could also be other adverse volatiles that drive females away?
L325: passive-active grammar error
L326: Here and in other places in the text, the use of spacing between the (+) or (-) and the molecule is inconsistent.
L363: Redundant sentence
L376-378: Again repetitive.
Discussion in general: A few things have been hinted at in the introduction and discussion that have not been properly explained, adding those would give the reader a better understanding about the significance of the research presented here.
- How could those attractants be used in an applied context to fight S. frugiperda infestations?
- Is there a tradeoff for S.frugiperda placing their eggs on the smallest plants, e.g. more competition for very limited resources? Could these plants even sustain the larvae, or would they then have to move on to the next plant?
- Is it common that these moths encounter maize in more than one growth state so that they would have to choose between several plants when depositing eggs?
Comments on the Quality of English Language
Overall the text is written in good english and is easy to read. I found a few flaws, probably translation errors for specific terms and in the captions/axis titles of graphics.
Author Response
Reviewer 2
Comments 1: L58-59: The growth stages could be better explained/defined
Response 1: Thank you for pointing this out. We explained the growth stages in lines 95-101.
Comments 2: L100: 75 Degrees.
Response 2: Thank you for pointing this out. We agree with this comment. We have added the sign of degree in line 110.
Comments 3: L114: While the meaning of the sentence is clear, could another word be used instead of baked?
Response 3: Thank you for pointing this out. We agree with this comment. We have replaced baked with heat-sterilized in lines 124-125.
Comments 4: L120: Could ‘coated’ be used here instead of ‘impregnated’?
Response 4: Thank you for pointing this out. We have modified 'impregnated' to 'coated' in line 130.
Comments 5: L121-122: How many insects were used in total and how many in each pairwise comparison?
Response 5: Thank you for pointing this out. We have added the required information about how many insects were used in total and for each pairwise comparison, in lines 132-135. Moreover, we have added the total number of test insects for each group of experiments in Figures 1 and 2.
Comments 6: L124: The method could be explained more precisely, how can each volatile have been tested 10 times, when 2 stages have been tested only once, and two others have been tested two times?
Response 6: Thank you for pointing this out. Each pairwise bidirectional volatile selection test was replicated 10 times per growth stage. For example, in the bidirectional selection experiment of FS and MS maize volatile substances, we placed 10 insects in the Y-tube each time and repeated it 10 times. We have provided detailed experimental procedures in lines 132-135. In order to unify the methods throughout the manuscript, we also modified Figure 3 in line 253.
Comments 7: L134: Newly emerged individuals
Response 7: Thank you for pointing this out. We have added the word ‘individuals’ to make the sentence expression more complete in line 149.
Comments 8: L135-136: A timeframe could be provided here
Response 8: Thank you for pointing this out. We conducted the experiment by placing fresh adults and leaves at 08:00 and recording results after 24 hours (at 08:00 the next day). We have supplemented the timeframe in lines 152-154.
Comments 9: L163: Is this supposed to mean females?
Response 9: Thank you for pointing this out. It does refer to the female here. We have modified it to females in line 180.
Comments 10: L199: Why was there no blank control in this experiment? Figure 9 shows controls for a different experiment.
Response 10: Thank you for pointing this out. Our preliminary tests showed that volatiles from maize at all growth stages showed significantly attractive effect to adult S. frugiperda compared to the blank control. Since it was impossible to determine their preference for specific growth stages by the comparison with blank control. Therefore, no blank control was in this experiment. We explained this in lines 220-223.
Comments 11: L201: What was the reason that only adjacent growth stages were compared and not all stages against all others, e.g. MS vs. SS
Response 11: Thank you for pointing this out. Previous investigation found that the sowing time was not uniform in maize production. Meanwhile, the differences in soil conditions lead to variations in the emergence time of seedlings. This will inevitably lead to adjacent growth stages in adjacent fields during the maize cultivation, thus the selection between adjacent growth stages by S. frugiperda is an important issue. Obviously, sowing time will not vary greatly, moreover comparisons between adjacent growth stages already showed a preference for young leaves. Thus we did not compare all stages against all others, e.g. MS vs. SS. We have supplemented it in lines 377-383.
Comments 12: L202-204: The number of insects that decided on one or the other side does not add up to groups of 10 insects nor does it add up to 10 insects tested 10 times. Also, the number of insects varies for each comparison. If a considerable number of insects did not decide or had to be exempt for some reason, this needs to be shown in the results.
Response 12: Thank you for pointing this out. Some adult insects did not make a selection of the odor sources in this experiment. We have added detailed methods in lines 132–135. Additionally, the total sample size for each experimental group has been included in Figures 1 and 2.
Comments 13: L206: The X-axis title is misleading/translated wrong – this also applies to the following similar graphics. For all graphics, one side of the bars could be a black outline and white inside to better contrast the two, especially for the graphs in figure 4.
Response 13: Thank you for pointing this out. We have modified the X-axis titles of Figures 1 and 2 in lines 231 and 238. We changed the inside fill of all the figures in the text to white to better contrast the two.
Comments 14: L207: n should be indicated in the caption (in all graphics). Captions for figures 1 and 2 are not concise.
Response 14: Thank you for pointing this out. The total sample size for each experimental group has been included in Figures 1 and 2. And the captions of Figure 1 and Figure 2 have been modified in lines 232-234 and 239-241.
Comments 15: L211: The bars do not need to be elevated compared to figure 1.
Response 15: Thank you for pointing this out. We have modified the bars in Figures 2 and 3 in lines 238 and 253.
Comments 16: L244: Figure A Y-Axis title is inconsistent with other graphs using percentages (F and H). One of the bars in the graphs could be white with black outline to better contrast the graphs, compared to two different shades of grey. Graph D Y-Axis title could be more precise in the graph and caption – worm weight- when other graphs say larval. Graph F uses a scale of 0.0-1.0 for percentage values when other graphs use 0-100, was the emergence rate 1 %?
Response 16: Thank you for pointing this out. We modify the Y-axis title of Figure 4-A to survival rate (%). We changed the inside fill of all the figures in the text to white to better contrast the two. We have modified the Y-axis title and caption in Figure 4-D. We have unified the representation of percentages in Figure 4.
Comments 17: L255: Can an x-axis be added to this graphic?
Response 17: Thank you for pointing this out. We added the axes to Figure 5 in line 285.
Comments 18: L266-269: Could other molecules have increased in percentage, adding to the recorded effect?
Response 18: Thank you for pointing this out. In GCMS detection, we found relative contents of some volatiles did not increased, for example the content of 3-Hexanol (retention time: 6.05 min) was 0.003%, 0.020%, 0.017% and 0.033% in SS, STS, FS and MS stages. It was the same with 1-methyl-cyclopentanol (retention time: 5.86 min). Please see Supplementary File Table 1-4.
Comments 19: L280: Is there an X-Axis to this graphic? Is there an explanation to what the red bars in the graphic are supposed to show the reader?
Response 19: Thank you for pointing this out. This graph has no X-axis. We refer to this article "Molecular Basis of Alarm Pheromone Detection in Aphids". The red bars indicate the position of the stimulus. We have provided a supplementary explanation of the red bars in line 316.
Comments 20: L293: Figure 8 X-Axis titles are inconsistent (spacing).
Response 20: Thank you for pointing this out. We have modified Figure 8 in line 330.
Comments 21: L294: a and b do not show up in the graphics. It is confusing to talk about graph a and graph b when both graphs contain ORN a and b. The term ORN is not explained in the text.
Response 21: Thank you for pointing this out. Neuron a and neuron b are judged based on the difference in the amplitude of the action potential (Figure 7). We have supplemented a detailed explanation of ORN in lines 310-313.
Comments 22: L298: the sentence is unnecessary, the previous chapters showed this.
Response 22: Thank you for pointing this out. We deleted the unnecessary sentence in line 336.
Comments 23: L306: Figure 9 – the term CK is not explained in the text
Response 23: Thank you for pointing this out. Liquid paraffin was used as the control (CK) in line 208.
Comments 24: L311: repetitive
Response 24: Thank you for pointing this out. We have deleted the repetitive sentence in line 350.
Comments 25: L314-315: is there any indication that there could also be other adverse volatiles that drive females away?
Response 25: Thank you for pointing this out. We believed the presence of repellent volatiles, but none were detected in our study.
Comments 26: L325: passive-active grammar error
Response 26: Thank you for pointing this out. We modified the tenses in line 366-372.
Comments 27: L326: Here and in other places in the text, the use of spacing between the (+) or (-) and the molecule is inconsistent.
Response 27: Thank you for pointing this out. We have unified the spacing in lines 359-366.
Comments 28: L363: Redundant sentence
Response 28: Thank you for pointing this out. We have deleted the repetitive sentence in line 424.
Comments 29: L376-378: Again repetitive.
Response 29: Thank you for pointing this out. We have revised the final conclusion in lines 438-442.
Comments 30: How could those attractants be used in an applied context to fight S. frugiperda infestations?
Response 30: Thank you for pointing this out. We propose that these plant volatiles can be developed into traps for field trapping. They may be used in combination with repellents as part of a push-pull strategy. Additionally, exploring the integration of these food attractants with sex attractants or visual cues could lead to more effective strategies for monitoring and controlling S. frugiperda populations. We provided supplementary explanations in lines 423 and 432.
Comments 31: Is there a tradeoff for S.frugiperda placing their eggs on the smallest plants, e.g. more competition for very limited resources? Could these plants even sustain the larvae, or would they then have to move on to the next plant?
Response 31: Thank you for pointing this out. When female insects lay eggs on the tender growth stages like SS stage, the larvae may face limited food resources due to the small leaf areas. The larvae will spread to adjacent plants through silk-spitting behavior. However, due to their limited mobility, larval survival ultimately depends on the oviposition sites selected by females. The selection for oviposition site of S. frugiperda is a trade-off between the quantity and quality of food. Our study had found that although young leaves may lead to scare food resources, females still preferred to choose high-quality food sources such as young leaves with rich nutrients and low levels of defensive secondary metabolites insect-resistant metabolites. We have added it in lines 383-393.
Comments 32: Is it common that these moths encounter maize in more than one growth state so that they would have to choose between several plants when depositing eggs?
Response 32: Thank you for pointing this out. We think this is a common phenomenon. Previous investigation found that the sowing time was not uniform in maize production. Meanwhile, the differences in soil conditions lead to variations in the emergence time of seedlings. This will inevitably lead to different growth stages in adjacent fields during the maize cultivation, thus these moths will encounter maize in more than one growth state. We have supplemented it in lines 377-383.
Round 2
Reviewer 2 Report
Comments and Suggestions for Authors
Dear Authors,
Thank you for responding to my comments. Your responsed have answered my questions and provided insight into your experiment. Still, I would like to see one more issue being adressed before I think this article is ready for publication.
Based on the information now provided, it seems that for the behavioral bioassay, most of the tested individuals did not pick any side, in most cases more than 50 % did not venture to any of the presented volatiles.
I am no expert for this kind of experimental approach, but I think this should be noted or explained, and that is also why I asked about a control, to see how common a non-response is. While I know that adding this now is probably not possible, I would like to see an explanation or comparison to other similar experiments.
This being said, could you provide more information about the oviposition experiment? You mention the average number of eggs but not the number of females that moved to either side in the experiment. Mentioning the number of eggs per female or the number of females that chose either side would provide valuable information here.
If all females eventually moved to either side, the high number of non-responses in the previous experiment might just be related to the limited time given to the individuals to move.
Figure 8 is missing a denoting a or b, which is mentioned in the caption.
Comments on the Quality of English LanguageDear Authors, you adressed all of my comments sufficiently.
Still, I would like you to revisit the X-axis titles for figures 1-3 and 9.
I think the term "the selected number of insects" is incorrect. I would prefer "number of choices" or something similar for figures 1,2 and 9.
For figure 3, I think the Average number of eggs/Mean number of eggs would suffice, given that oviposition time was only 1 day anyways. If the oviposition time would have been longer, the other title would be appropriate
Author Response
Reviewer 2
Comments 1: Dear Authors,
Thank you for responding to my comments. Your responsed have answered my questions and provided insight into your experiment. Still, I would like to see one more issue being adressed before I think this article is ready for publication.
Based on the information now provided, it seems that for the behavioral bioassay, most of the tested individuals did not pick any side, in most cases more than 50 % did not venture to any of the presented volatiles.
I am no expert for this kind of experimental approach, but I think this should be noted or explained, and that is also why I asked about a control, to see how common a non-response is. While I know that adding this now is probably not possible, I would like to see an explanation or comparison to other similar experiments.
This being said, could you provide more information about the oviposition experiment? You mention the average number of eggs but not the number of females that moved to either side in the experiment. Mentioning the number of eggs per female or the number of females that chose either side would provide valuable information here.
If all females eventually moved to either side, the high number of non-responses in the previous experiment might just be related to the limited time given to the individuals to move.
Response 1: Thank you for pointing this out. We think this is a very good comment which prompt us to think deeply about our data. We find that it is a common phenomenon for the response rate of olfactory recognition to be lower than 50% by the academic communication with other researchers, for example data showed in the article titled "Behavioral responses of Diaphorina citri to host plant volatiles in multiple-choice olfactometers are affected in interpretable ways by effects of background colors and airflows". https://doi.org/10.1371/journal.pone.0235630. Of course, there are some papers reporting a relatively high response rate. We think this might be due to the different patterns of host recognition by pests. An interesting results is that most of the adults made their choices in the experiment of oviposition selectivity, except for two that died in the experiment (data was showed in line 267-271 and Figure 3). The possible reason might be that in the experiment of oviposition preference, the host plants were real and not merely the volatile substances simulating their odors. Obviously, the real host plants provided abundant cues, thus demonstrating a stronger attraction than the sole olfactory cues. This result indicated that there might be other factors besides olfactory cues influencing the host recognition of S. frugiperda, which can be further studied in the future. Therefore, the comprehensive utilization of multiple clues can improve the trapping efficiency. We added in the discussion that in lines 455-468.
Comments 2: Figure 8 is missing a denoting a or b, which is mentioned in the caption.
Response 2: Thank you for pointing this out. We have modified Figure 8 and its caption.
Response to Comments on the Quality of English Language
Point 1: Dear Authors, you adressed all of my comments sufficiently. Still, I would like you to revisit the X-axis titles for figures 1-3 and 9. I think the term "the selected number of insects" is incorrect. I would prefer "number of choices" or something similar for figures 1,2 and 9.
Response 1: Thank you for pointing this out. We have modified the X-axis titles for Figures 1, 2 and 9 to "Number of choices" .
Point 2: For figure 3, I think the Average number of eggs/Mean number of eggs would suffice, given that oviposition time was only 1 day anyways. If the oviposition time would have been longer, the other title would be appropriate.
Response 2: Thank you for pointing this out. We have modified the X-axis titles for Figure 3 to "Average number of eggs".